# Evidence of *Dirofilaria immitis* in Felids in North-Eastern Italy

**DOI:** 10.3390/pathogens11101216

**Published:** 2022-10-20

**Authors:** Marika Grillini, Antonio Frangipane di Regalbono, Cinzia Tessarin, Paola Beraldo, Rudi Cassini, Erica Marchiori, Giulia Simonato

**Affiliations:** 1Department of Animal Medicine, Production and Health, University of Padua, 35020 Legnaro, Italy; 2Department of Agricultural, Food, Environmental and Animal Sciences, University of Udine, 33100 Udine, Italy

**Keywords:** *Dirofilaria immitis*, heartworm, cat, wildcat, felid, North-Eastern Italy

## Abstract

*Dirofilaria immitis* is a mosquito-borne nematode, causing heartworm (HW) disease in wild and domestic canids. HW can also affect felids with different clinical patterns from asymptomatic pictures to sudden death, making the monitoring and diagnosis complicated. Canine HW is endemic in North-eastern Italy; however, very little information has been recorded for felids. This study aims to provide new information on HW in felids in North-eastern Italy. Two hundred and six domestic cats from Veneto, Friuli-Venezia Giulia, Trentino Alto-Adige regions (North-eastern Italy), nine captive felids from zoological parks from Veneto, and nineteen European wildcats from Friuli Venezia Giulia were recruited. Sera/plasma was analysed for the detection of anti-HW antibodies (Ab) and HW antigens (Ag); positive blood samples were molecularly analysed, targeting the HW DNA (5S-rRNA gene). Twelve out of two hundred and six (5.8%) cats presented with Ab, and three out of two hundred and six (1.5%) presented with Ag, mainly those from the Veneto region, already known as a canine HW-endemic area. Among Ab-positive cats, two were from Belluno, a mountain province previously considered free, suggesting the expansion of HW into the northern areas. No cats were positive for both Ab and Ag. Three out of nineteen (15.8%) wildcats were Ag-positive, constituting the first HW report in Italy. No captive felids were positive. *Dirofilaria immitis* DNA was not amplified in positive samples, suggesting the low sensitivity of PCR on blood. This study provides new data on the occurrence of HW in domestic cats and wildcats in North-eastern Italy.

## 1. Introduction

*Dirofilaria immitis* is a nematode endemic present in many parts of the world, from European countries to the northern states of America and in South-east Asia. Additionally, an increasing frequency has been reported in the African regions [1].

In Europe, *D. immitis* is mostly endemic in the southern countries such as Spain and the Canary Islands [2,3,4,5], Portugal [6], France [7], and Greece [8,9]; as for central and eastern countries, the nematode has been detected in both dogs and cats from Romania [10,11,12,13], Czech Republic, Slovenia, Bulgaria [14], and Austria [15].

In Italy, northern regions, such as, for instance, Veneto, Friuli Venezia Giulia, Emilia Romagna, and Piedmont e Lombardy, are reported as hyperendemic. Indeed, the largest endemic area in Europe is along the Po River Valley [16]. *Dirofilaria immitis* is generally reported as being considerably distributed in the northern and central Italian regions [17].

Nevertheless, in Italy and in other European regions, the nematode is expanding its geographical range, affecting previously free areas [11,16,18].

Recently, some authors described an increase in *D. immitis* in Central and Southern Italy, throughout the Tuscany and Umbria regions [19,20] and in Sardinia and the Sicily islands [21]. Finally, *D. immitis* has been recently detected in the Calabria and Apulia regions [19,22,23].

The definitive hosts for this heartworm (HW) are dogs (*Canis lupus familiaris)* and other canids such as wolf (*Canis lupus*), fox (*Vulpes vulpes)*, and European jackal (*Canis aureus*). Nevertheless, a large percentage of other species could be infected by this species, such as, for instance, the domestic cat (*Felis silvestris catus)*, ferret (*Mustela putorius furo*), and coypu (*Myocastor coypus*); however, these, unlike canids, do not act as a reservoir for this parasite [1].

Focusing on felids, *D. immitis* infestation is reported with much more complexity than in dogs due to the fact that felines are vulnerable hosts but not the HWs’ favourite ones [24]. Indeed, the prepatent period in felines is extended up to 9 months compared to canids [25]. Moreover, adult worms are lower in number, with a shorter lifecycle, sometimes reaching more ectopic locations than the target ones, and microfilaremia is rarely present, with a scant burden [24,26].

Genchi et al. [17] recently reported data focusing on HW disease in domestic cats, collected through a national questionnaire sent to Italian veterinarian practitioners. Between 2017 and 2018, one or two clinical cases of HW disease/year were diagnosed in the provinces of Veneto, Emilia Romagna, Lombardy, Piedmont, Tuscany, and Sardinia, and more than two clinical cases were diagnosed in the Lombardy and Sardinia provinces.

Even if data on the prevalence of HW in dogs are not fully updated in North-eastern Italy, the circulation of the parasite is well known, and the adoption of preventative measures in dogs is widespread during the period of mosquito activity (i.e., from late spring to late autumn). On the other hand, feline HW disease is not fully understood, and vet awareness in endemic areas is still scant and data on HW prevalence rates in felids are still lacking.

Nevertheless, it is important to consider that the HW disease in cats may potentially occur wherever infested dogs and competent vectors are present in the same context [27]. The hypothetical prevalence of *D. immitis* in cats is around 9–18% of that in dogs in the same area [28]. Although North-eastern Italy has experienced the presence of *D. immitis* for a long time [29], information on HW in felids is still lacking. Some invasive competent diurnal mosquitoes were recently introduced [30,31], exposing dogs and cats to a major risk of infestation. The aim of this study is to provide new information on *D. immitis* circulation in different species of felines.

## 2. Results

### 2.1. Feline Population

Overall, 234 felids (i.e., 206 domestic cats, 9 captive exotic felids, and 19 European wildcats) were included in the study. Among the cats, 146 (70.9%) were from Veneto (Site 1), 40 (19.4%) were from Friuli Venezia Giulia (Site 2), and 20 (9.7%) were from Trentino Alto-Adige (Site 3); 131 (63.6%) and 75 (36.4%) were owned and stray cats, respectively. Most of the cats (n = 145, 70.4%) were recorded as having an outdoor lifestyle, and the rest (n = 61, 29.6%) had an indoor lifestyle. Recruited subjects are almost equally distributed among the sex, provenance, and age classes. Individual data are detailed in Table 1.

The exotic felids included three tigers (*Panthera tigris*), two lions (*Panthera leo*), three leopards (*Panthera pardus*), and one caracal *(Caracal caracal*) from zoological parks located in Site 1. Among them, five were male and four were female, and all were aged between 4 and 20 years.

The collected European wildcats (*Felis silvestris silvestris*) were road-killed in Site 2 (i.e., the Trieste (TS), Udine (UD), and Pordenone (PN) provinces). The wild felids included 10 (52.6%) males and 9 (47.4%) females. An estimated age based on teeth evaluation classified the animals as adults if older than one year (17/19, 89.5%) or sub-adults if younger (2/19, 10.5%).

### 2.2. Laboratory Analysis and Geographical Distribution

Twelve out of two hundred and six (5.8%) domestic cats presented positive for anti-HW Ab, and three out of two hundred and six (1.5%) presented positive for HW Ag. None of them presented positive for both tests. Among the three Ag-positive animals, one was positive only after the heat treatment of the serum. Positive cats (n = 15/206, 7.3%) were mostly asymptomatic. Nevertheless, 3/15 (20.0%) positive cats presented cardio-respiratory signs: one Ab-positive cat showed a light cardiac murmur, one Ag-positive cat a mistral regurgitation, and another Ag-positive one wheezing and shallow breathing.

Eleven out of twelve Ab- and three out of three Ag-positive cats came from Site 1 and only one Ab-positive cat was from the Trieste (TS) province in Site 2.

Three out of nineteen (15.8%) wildcats, including one adult and two sub-adults, revealed slight positivity to antigens, whereas none were positive for Ab. Among them, two came from Udine (UD) and another from Trieste (TS).

No exotic felids tested positive at serological tests.

Serological positive samples were all negative after the molecular analysis.

*Dirofilaria immitis* is widely distributed in Site 1 and sporadically present in Site 2 (Figure 1).

The Ab-positive cats from the Belluno (BL) province (Site 1) were asymptomatic and came from a street colony without a history of travelling. No cats were recruited from the Rovigo (RO, Site 1), Gorizia (GO, Site 2), and Bolzano (BZ, Site 3) provinces; thus, no information regarding the presence of *D. immitis* is available for these areas. On the contrary, the Trento (TN, Site 3) province did not record any positivity.

Individual data of Ab- and Ag-positive cats are reported in Table 2.

No significant difference in prevalence was observed among the groups, even if stray cats showed a higher rate of positivity for Ab anti-*D. immitis* as well as cats with an outdoor lifestyle and younger cats with an age between 12 and 36 months, whereas positivity rates for Ag seem to proportionally decrease with an increase in age.

## 3. Discussion

In the last years, several drivers have contributed to modifying the epidemiological distribution of parasites in Europe. For example, climate change is facilitating an increase in vectors’ spread and activity [32], and globalization in terms of animal and goods movement across the world is allowing the introduction of arthropod vectors and pathogens in new territories [33,34]. Moreover, the decimation of ecological wildlife niches is responsible for the approaching of wild animals to urban contexts, exposing domestic animals to new pathogens [35], and the pets’ movement is facilitating the spread of pathogens in new areas [36]. These factors have expanded the influence of parasites including cardio-pulmonary nematodes from areas already endemic to regions previously described as free [35].

Specifically, environmental factors (e.g., temperature, humidity, vegetation, etc.), the density of competent mosquito populations, the new introduction of invasive competent mosquitoes [37,38], the presence of the main reservoirs of *D. immitis* (i.e., wild and domestic canids), together with the movement of microfilaremic individuals, plays an important role in the increased risk of exposure, even for the feline population.

Canids remain the favourite definitive hosts of *D. immitis* and contribute to the maintenance of the parasite in the domestic and sylvatic cycles. In addition, in some mosquito species (i.e., *Culex* spp. and *Aedes* spp.), the main competent vectors of *D. immitis* are very common in urban areas and feed on both dogs and cats with no preferences [1].

Generally, cats living in endemic and hyperendemic areas for canine HW disease should be considered at risk [39], and the prevalence rates in the feline species are considered to fluctuate from 9 to 18% of that in the canine population in the same area [28]. In this study, the provenance of stray cats from neighbouring areas and the data collected from cats’ owners allow us to rule out with a good level of certainty the introduction of domestic cats from endemic regions.

Unfortunately, HW disease is difficult to diagnose in felines due to the fact that it is much more elusive and because infection leads to unpredictable effects in this host. As previously reported, felines are usually asymptomatic or paucisymptomatic, and infection can also lead to sudden death [9,40,41]. Moreover, no single test is able to detect the presence of *D. immitis* in all its stages [1,42]. Indeed, usually, microfilaremia is not frequent and, if present, has a scant burden [24,26] due to the fact that heartworms tend to die before reaching adult stage.

Consequently, more diagnostic methods should be combined to confirm the exposure and/or the infestation, always considering the limits of available serological tests [25,28].

In this study, the use of two different tests was planned in order to detect the presence of anti-*D. immitis* Ab and/or *D. immitis* Ag. No cats tested positive for both tests, and this could be due to the production of Ab anti-*D. immitis* in the first stage of parasitosis when the humoral immune response reacts to larval stages developing in the host’s tissues, whereas antigens are present in the final stage only when the nematodes become sexually mature adults. Antibodies are early detectable at 3–4 months post-infection, whereas Ags are present for around 6–8 months post-infection [25]. Our findings (5.8% prevalence for Ab, 1.5% for Ag) confirm that the Ab prevalence rates are usually higher than the Ag values because, in cats, *D. immitis* immature larvae have more difficulty reaching the adult stage, and because circulating immune complexes, often present in cats, act to mask the Ag.

The presence of circulating anti-*D. immitis* Ab in felines is considered a useful indication in the diagnostic process. Indeed, it provides early information of dirofilariosis onset, considering that Abs develop within 4 months of the infestation. This condition may permit the identification of animals potentially infested and can be proceeded with further diagnostic tests to confirm the risk of the subsequent onset of the disease. However, we should be aware that the Ab presence may be simply an indication that cats came in contact with the parasite; alternatively, it may mean that, although infected, they will not necessarily develop the disease. In fact, felines seem to tolerate the infestation well, sometimes with no clinical signs or with signs that occur only transiently [25]. Additionally, the HW disease often has a self-limiting course in felids with a spontaneous resolution due to the natural death of the parasites [1,28]. In general, the specificity of antibody tests can drop significantly due to possible cross-reactivity with other parasites. Regardless, it can be assumed that the tests commercially available to detect *D. immitis* antibodies rarely cross-react with gastro-intestinal parasites [43]. Moreover, no autochthonous cases of *D. repens* infection have been reported in the investigated regions [17].

*Dirofilaria immitis* DNA could not be molecularly detected in any positive samples, suggesting the low sensitivity of PCR on the blood matrix. Indeed, molecular procedures are frequently used for nematode identification in dog blood samples [44,45,46,47], whereas, in cats, they are just marginally applied [48,49].

Most of the Ab-positive cats had an outdoor lifestyle (6.9%), and this condition definitely exposes cats to vectors’ action night and day, as well as to wildcats. Nonetheless, a lower percentage (3.3%) of Ab-positive cats were recorded as having an indoor lifestyle. An indoor lifestyle can only partially protect them from vectors’ bites since some mosquito species are attracted inside human dwellings [50] and are active both night and day.

In Europe, few reports of feline HW clinical cases have been described in southern Romania [13] and in Austria [15], and only a few serosurveys in Spain [3,4] and in Greece [9] were reported. Seroprevalence data registered in our study agrees with previous studies in other endemic regions of Spain; indeed, Ab and Ag seropositivity rates in North-eastern Italy were 5.8% and 1.5%, respectively, compared to those reported in Madrid (7.30% and 0.20%) [4] and Barcelona (11.47% and 0.26%) [3].

In Italy, the Po Valley is a hyperendemic area for canine HW disease and thus for cardio-pulmonary dirofilariosis in cats [16], as confirmed by the number of Ab- (11/146, 7.5%) and Ag-positive (3/146, 2.0%) samples registered in cats from the Veneto region, followed by cats from Friuli Venezia Giulia, where 1/40 (2.5%) was Ab-positive and 3/19 (15.8%) wildcats were Ag-positive. No cat from the Trentino Alto-Adige region was Ab- or Ag-positive, as has been already described in another study [17], suggesting that cats are not exposed to the risk of infestation, probably due to the climatic conditions that are not yet suitable for the establishment of a *D. immitis* lifecycle.

Feline dirofilariosis was diagnosed in 4.8% of Italian facilities in 2018 [17]. In North-eastern Italy, only a few cases of infested cats were described in the Treviso (TV) province, and no information was reported from the Belluno (BL) province.

In this study, several provinces proved to be positive for the circulation of *D. immitis* in cats. Particularly, two cats showed positivity to anti-*D. immitis* antibodies in Belluno, which represents an area recently colonized by a new mosquito species [30,31,33]. This is worthy of note due to the fact that, to the best of the authors’ knowledge, this is the first description in this province that has the highest mean altitude (390 masl) among the other provinces included in the study. This altitude is commonly not suitable for mosquitoes, even if, since 2011, a new species (i.e., *Aedes koreicus*)-competent vector for *D. immitis* with a particular resistance to colder environmental temperatures was described [30]. It may suggest that, hitherto, *Dirofilaria*-free areas are to be considered at potential risk of spread.

Focusing on wildcats, to the best of the authors’ knowledge, it is the first time that wildcats have been investigated for heartworm in Italy.

Felines do not properly act as reservoirs, since the parasites rarely reach the sexually mature adult stage that produces microfilariae. Regardless, in this study, the presence of *D. immitis* Ag in some felids supports the presence of mature adult worms, in contrast with the necropsy not showing their presence. This result could be due to the very few adult worms (as usual in felines) reaching the target organs investigated at the necropsy, and/or to the deteriorating effects on worms’ tissues due to the long freezing storage (more than 1 year).

Exotic felids were negative on both tests. Indeed, these felines come from zoological gardens that are used to adopt prevention programmes based on oral ivermectin administration. This could reinforce our findings and confirm the efficacy of chemoprophylaxis in captive felids which are housed outdoors in endemic HW areas since they were all found to be negative; however, the number of sampled animals was low.

The antiparasitic treatment is crucial in cats as well as in dogs. As documented by Genchi et al. [17], more than half of veterinary practitioners do not recommend HW prophylaxis for cats. This aspect agrees with our data regarding the preventative measures adopted by cat owners. In this study, it is not possible to affirm that cats with preventive measures were protected for their whole life until blood sampling. Only 16.5% of cat owners adopted formulations for endo and/or ectoparasites treatment, not aware that the molecules were also effective against the larval stages of *D. immitis*. Unfortunately, the irregular regimen of treatments or the administration only when necessary for other parasitosis makes the preventative measures against dirofilariosis partially ineffective.

A registered adulticide molecule is not available for cats, and it is considered to be a last-resort medical treatment for those with uncontrolled clinical signs after empirical corticosteroid therapy [25]. To date, data on melarsomine dihydrochloride in cats are scant and its use is not recommended [25]. Moreover, preliminary studies suggest that melarsomine at a regimen of 3.5 mg/kg is toxic for cats [51,52]. Ivermectin in monthly doses seems to reduce the number of adult worms by 65% compared to untreated cats; on the other hand, its use could lead to anaphylactic reactions due to the death of heartworms. Currently, there are no studies showing that adulticide therapy is competent enough to increase the rate of survival of cats infected by adult worms [25].

The adoption of several topical and/or oral formulations available on the market is the only suggested way to protect cats from mosquitoes’ bites and guarantee cat health [17]. Since mosquitoes are anthropophilic, this treatment should also be provided for cats which have an indoor habit. Indeed, an indoor lifestyle can partially limit their exposure but cannot avoid the risk of infestation [49,53].

This study proved the circulation of *D. immitis* in domestic cats and, for the first time, in wildcats from several provinces of North-eastern Italy. As a consequence, it has highlighted the importance of taking regular preventative measures for cats living in endemic or hyper-endemic areas, along the need to include HW disease in the differential diagnosis. In clinical practice, the elusive picture and the unpredictable effects of HW infection in cats leads to the need for a combination of more diagnostic methods to confirm the exposure and/or the infestation. Considering the limits of available serological tests, the possibility to employ imaging diagnostics to determine the presence of heartworms is recommended.

## 4. Materials and Methods

### 4.1. Blood Collection, Analysis, and DNA Extraction

Blood samples from different felid species from North-eastern Italy were sampled from October 2019 to March 2022. Among felid species, domestic cats, captive exotic felids (i.e., tigers, lions, leopards, caracals) from zoological parks, and wild European wildcats were included. All felids were exposed to at least one season at risk of arthropod vectors’ activity.

The investigated areas included the Veneto (Site 1), Friuli Venezia Giulia (Site 2), and Trentino Alto-Adige (Site 3) regions (Figure 2). Mean altitudes of the different provinces are represented in Figure 1.

Blood samples (i.e., entire blood and blood in k3EDTA) were collected from cats and exotic captive felids during routine clinical visits and/or following surgical procedures (not depending on this research study) in veterinary clinics. Individual data upon provenance (Site 1, Site 2, and Site 3), sex, and age classes (<12 months, 12–36 months, >36 months), management (owned, stray cats, captive, wild), lifestyle (indoor, outdoor, mixed), and clinical signs (cardio-respiratory alterations) were recorded.

The included European wildcats were found road-killed in the territory and kept frozen until post-mortem examination. During necropsy, organs were observed to reveal the presence of *D. immitis* pre-adults and adults. Heart clots and/or uncoagulated blood (when present) were sampled for serological and molecular investigations.

Sera were analysed for the detection of antibodies (Ab) anti-*D. immitis* by one-step lateral flow immunoassay Solo Step^®^ FH (HESKA^®^ Corporation, Loveland, Colorado, USA) and antigens (Ag) of *D. immitis* by enzyme immunoassay PetChek^®^ HTWM PF, (IDEXX Laboratories, Westbrook, Maine, USA) according to manufacturer’s instructions.

For the Ag test, the domestic cats’ sera were tested before and after heat treatment (104°C for 10 min, then centrifuged at 13,000 rpm) to avoid the potential interference of the antigen–antibody complex and to guarantee more accurate results in the antigen test [54]. The serum of 12 domestic cats was insufficient and was not analysed for antigens. For the same reason, the serum of exotic felids and wildcats was tested only at room temperature.

### 4.2. Molecular Analysis and Sequencing

Samples tested positive for serological investigations (i.e., Ab and/or Ag positive) were subsequently analysed by molecular method to detect the DNA of *D. immitis*.

The DNA extraction kit NucleoSpin^®^Tissue (Macherey-Nagel, Düren, Germany) was used on 200 μL of whole blood or clot. All procedures were performed according to the manufacturer’s instructions.

DNA was extracted from 200 μL of whole blood or blood clots with the NucleoSpin^®^Tissue kit (Macherey-Nagel, Düren, Germany) according to the manufacturer’s instructions.

PCR amplification of the ribosomal large subunit (5S) was carried out using the specific primers S2 (5′-GTTAAGCAACGTTGGGCCTGG-3′) and S16 (5′-TTGACAGATCGGACGAGATG-3′) [55] according to Favia et al. [56] with slight modifications: initial step at 95 °C for 5 min, followed by 35 cycles of denaturation step at 95 °C for 30 sec, annealing step at 56.5 °C for 30 sec, extension step at 72 °C for 30 sec, and final extension cycle at 72 °C for 2 min. Positive (i.e., DNA of *D. immitis*) and negative controls were included in each PCR reaction.

### 4.3. Data Analysis

In order to evaluate differences in infection rates among the subgroups of the investigated domestic cat population, a statistical evaluation was performed by means of the Pearson chi-square test or the Fisher exact test, if appropriate, using SPSS for Windows, version 27.0. The factors taken into consideration were sex (i.e., males, females), age classes (i.e., <12 months, 12–36 months, >36 months), region and province of provenance (i.e., Site 1, Site 2, Site 3), lifestyle (i.e., indoor, outdoor), management (i.e., owned, stray cat), and presence of clinical signs (i.e., cardio-respiratory signs).

Due to the low number of samples, captive and wild felids were not included in this analysis.

## Figures and Tables

**Figure 1 pathogens-11-01216-f001:**
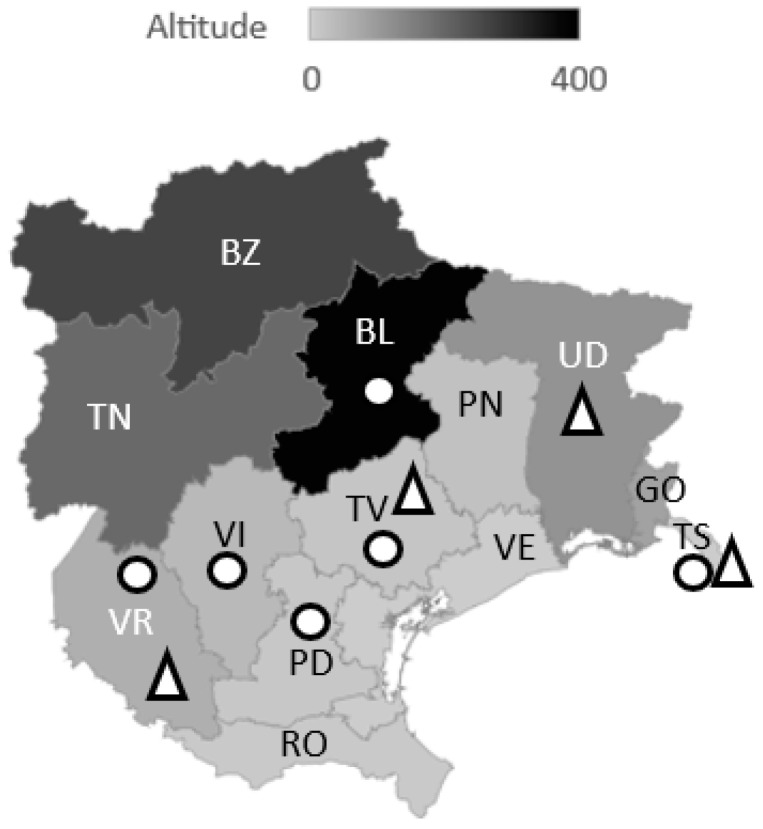
Altitude map showing Italian provinces with felines positive to anti-*D. immitis* antibodies (○) and to *D. immitis* antigens (∆); Site 1 provinces: BL, Belluno; PD, Padova; RO, Rovigo; TN, Trento; TV, Treviso; VE, Venezia; VI, Vicenza; VR, Verona; Site 2 provinces: GO, Gorizia; PN, Pordenone; TS, Trieste; UD, Udine; Site 3 provinces: BZ, Bolzano; TN, Trento.

**Figure 2 pathogens-11-01216-f002:**
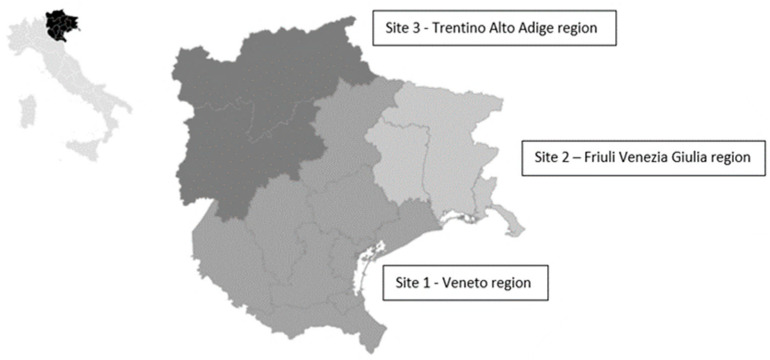
Map depicting investigated regions of North-eastern Italy.

**Table 1 pathogens-11-01216-t001:** Descriptions of individual data of domestic cats.

		Site 1 n (%)	Site 2 n (%)	Site 3 n (%)	Total n (%)
Sex	M	74 (50.7)	14 (35.0)	12 (60.0)	100 (48.5)
	F	70 (47.9)	26 (65.0)	8 (40.0)	104 (50.5)
	NR ^1^	2 (1.4)	0	0	2 (1.0)
Age classes (months)	<12	51 (34.9)	9 (22.5)	5 (25.0)	65 (31.6)
	12–36	41 (28.1)	17 (42.5)	8 (40.0)	66 (32.0)
	>36	43 (29.5)	12 (30.0)	7 (35.0)	62 (30.1)
	NR ^1^	11 (7.5)	2 (5.0)	0	13 (6.3)
Management	Owned	91 (62.3)	20 (50.0)	20 (100.0)	131 (63.6)
	Stray	55 (37.7)	20 (50.0)	0	75 (36.4)
Lifestyle	Indoor	42 (28.8)	12 (30.0)	7 (35.0)	61 (29.6)
	Outdoor	104 (71.2)	28 (70.0)	13 (65.0)	145 (70.4)
Cardio-respiratory Signs	Presence	8 (5.5)	1 (2.5)	1 (5.0)	10 (4.9)
	Absence	138 (94.5)	39 (97.5)	19 (95.0)	196 (95.1)
*Dirofilaria immitis* preventative measures	Presence	27 (18.5)	6 (15.0)	1 (5.0)	34 (16.5)
	Absence	108 (74.0)	33 (82.5)	19 (95.0)	160 (77.7)
	NR ^1^	11 (7.5)	1 (2.5)	0	12 (5.8)
Total		146 (70.9)	40 (19.4)	20 (9.7)	206 (100.0)

^1^ Not reported.

**Table 2 pathogens-11-01216-t002:** Individual data of domestic cats with anti-HW antibodies and HW antigens.

Factors	Variables	Tested N	*Dirofilaria immitis*	
			Ab+ n (%)	Ag+ n (%)
Provenance	Site 1	146	11 (7.5)	3 (2.1)
	Site 2	40	1 (2.5)	0 (0.0)
	Site 3	20	0 (0.0)	0 (0.0)
Sex	M	100	7 (7.0)	1 (1.0)
	F	104	5 (4.8)	2 (1.9)
	NR ^1^	2	0 (0.0)	0 (0.0)
Age classes (months)	<12	65	2 (3.1)	2 (3.1)
	12–36	66	5 (7.6)	1 (1.5)
	>36	62	4 (6.5)	0 (0.0)
	NR ^1^	13	1 (7.7)	0 (0.0)
Management	Owned	131	7 (5.3)	2 (1.5)
	Stray	75	5 (6.7)	1 (1.3)
Lifestyle	Indoor	61	2 (3.3)	1 (1.6)
	Outdoor	145	10 (6.9)	2 (1.4)
Cardiorespiratory signs	Presence	10	1 (10.0)	2 (20.0)
	Absence	196	11 (5.6)	1 (0.5)
*Dirofilaria immitis* preventative measures	Presence	34	3 (8.8)	2 (5.9)
	Absence	160	8 (5.0)	1 (0.6)
	NR ^1^	12	1 (8.3)	0 (0.0)
Total		206	12 (5.8)	3 (1.5)

^1^ Not reported.

## Data Availability

The authors declare that data are available upon request to the corresponding author, by email.

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
