# Peer review of "Evidence of Dirofilaria immitis in Felids in North-Eastern Italy"

_pathogens, 2022, doi:10.3390/pathogens11101216_

Round 1

Reviewer 1 Report

Review on the manuscript ID: pathogens-1916639

Title: Evidence of Dirofilaria immitis in Felids in North-Eastern Italy

By: Marika Grillini et al.

This is a very good contribution to the knowledge on feline HW (distribution, epidemiology) in Europe. An important part of Italy is investigated which is of major interest as HW might spread to the north within Europe.

Methods are reasonable and well discribed to evaluate the feline population.

Just a few remarks and suggestions:

Line 30: maybe include also North America

Line 73-77: long sentence - consider splitting it, as various information is included

Please comment on possible import of domestic cats from endemic regions - can you rule out such cases?

Please give a short comment on possible (or impossible) cross reactivity of antibody tests (e.g. sensitivity, specificity) in Dirofilaria repens infested cats

Regarding preventive measures: does it mean that cats with preventive measures were protected the whole life until blood sampling, or could there have been also some periods without preventive measures - in other words: does preventive measures or no preventive measures cover 100% of lifetime in your feline population?

Author Response

Dear Reviewer,

we are pleased to submit the revised version of “Evidence of Dirofilaria immitis in Felids in North-Eastern Italy”. We highly appreciate the reviewer’s comments on our manuscript. Here below a detailed response:

-Line 30: maybe include also North America

We have added the information as suggested (line 31)

- Line 73-77: long sentence - consider splitting it, as various information is included

We have split the sentence as suggested (lines 74-76)

-Please comment on possible import of domestic cats from endemic regions - can you rule out such cases?

We can state with sa good level of certainty that stray cats come from areas adjacent to the one under investigation. Regarding owned cats, the provenance reporting data allow us to exclude that they are from other endemic regions. We added a comment on that (lines 153-156)

- Please give a short comment on possible (or impossible) cross reactivity of antibody tests (e.g. sensitivity, specificity) in Dirofilaria repens infested cats.

The manufacturer of the test declares a sensitivity of 95%, and specificity of 99%. In dog, antibody tests have been abandoned because of the low specificity, due to the presence of antibodies for abortive infections or to possible cross-reactivity with other parasites. Anyway, it is assumed that antibody tests commercially available for cats have minimal cross-reactivity with gastro-intestinal parasites (Snyder et al, 2000, JAVMA 216, 693-700). Moreover, no autochthonous cases of D. repens infection have been reported in the investigated regions. We added a brief comment on that (lines 186-190)

- Regarding preventive measures: does it mean that cats with preventive measures were protected the whole life until blood sampling, or could there have been also some periods without preventive measures - in other words: does preventive measures or no preventive measures cover 100% of lifetime in your feline population?

Thank You. We added a brief comment on that (lines 247-248). Indeed, for the most part, the cats did not receive preventative treatments, and for cats that received them, we are not sure that the application/administration was regular until the blood sampling.

Reviewer 2 Report

Comments and Suggestions for Authors

Evidence of Dirofilaria immitis in Felids in North-Eastern Italy by M. Grillini, A. Frangipane di Regalbono , C. Tessarin, P. Beraldo, R. Cassini, E. Marchiori and G. Simonato. Authors sought for heartworm infection in 234 felids (i.e. 206 domestic cats, 9 captive exotic felids, 19 European wild- 80 cats) from North-Eastern Italy using different assays targeting the anti-HW antibodies and HW-antigens followed by molecular detection from positive samples. The technique employed to detect feline dirofilariosis is of particular interest. The experiments are well described, and the article is clearly written. It shows the current epidemiology of feline dirofilariosis in North-Eastern Italy. Therefore, the manuscript can be accepted in the type of Original Article, as it fits the purposes and aims of the journal, with minor revisions.

I don’t have particular concerns about the manuscript; however, some minor changes can improve the discussion and the overall value of the study. 

General comment for the authors:

Consider that the discussion is too long and there is many unnecessary information’s. Also, the authors should discuss the fact that feline disofilariosis is basically subjected to the host-immune response against the parasite, leading to parasite death before being adults, thus microfilaria and antigen production appear to be very rare. This statement should also be used to discuss the observed inconsistencies between the different diagnosis assays used by the authors. You can refer to McCall et al. (PMID: 18486691).

The article lacks the conclusion. 

Author Response

Dear Reviewer,

we are pleased to submit the revised version of “Evidence of Dirofilaria immitis in Felids in North-Eastern Italy”. We highly appreciate the reviewer’s comments on our manuscript. Here below a detailed response:

- Consider that the discussion is too long and there is many unnecessary information’s.

We appreciate the reviewer comment, but we also needed to address other reviewer’s suggestions involving new sentences in discussion. Anyway, we tried to meet the demand cutting out the unnecessary sentence “In 2019, Genchi et al. [17] …” (from line 215).

- Also, the authors should discuss the fact that feline dirofilariosis is basically subjected to the host-immune response against the parasite, leading to parasite death before being adults, thus microfilaria and antigen production appear to be very rare. This statement should also be used to discuss the observed inconsistencies between the different diagnosis assays used by the authors. You can refer to McCall et al. (PMID: 18486691).

We think that this information has been already reported in lines 164-165 and 233-234, but to highlight better what the reviewer suggests we added a more focused sentence at lines 161-163

- The article lacks the conclusion. 

Thank You. More detailed conclusions have been included in the last lines of the discussion (lines 267-274).